# Neuroprotective Potential of Glycyrrhizic Acid in Ischemic Stroke: Mechanisms and Therapeutic Prospects

**DOI:** 10.3390/ph17111493

**Published:** 2024-11-07

**Authors:** Yanwen Li, Juan Wu, Fang Du, Tao Tang, Jonathan Chee Woei Lim, Thilakavathy Karuppiah, Jiaxin Liu, Zhong Sun

**Affiliations:** 1Department of Basic Medicine, Medical School, Kunming University of Science and Technology, Kunming 650500, China; 20232136055@stu.kust.edu.cn (Y.L.); 20232136006@stu.kust.edu.cn (J.W.); 20202136050@stu.kust.edu.cn (F.D.); 20140072@kust.edu.cn (T.T.); 2Department of Medicine, Faculty of Medicine & Health Sciences, Universiti Putra Malaysia, UPM, Serdang 43400, Malaysia; cheewoei@upm.edu.my; 3Department of Biomedical Sciences, Faculty of Medicine & Health Sciences, Universiti Putra Malaysia, UPM, Serdang 43400, Malaysia; thilathy@upm.edu.my; 4Genetics and Regenerative Medicine Research Centre, Universiti Putra Malaysia, UPM, Serdang 43400, Malaysia

**Keywords:** glycyrrhizic acid, *Glycyrrhiza glabra*, ischemic stroke, cerebral ischemia-reperfusion injury, neuroprotection, HMGB1, oxidative stress, apoptosis, autophagy, Keap1/Nrf2

## Abstract

**Background/Objectives:** Ischemic stroke is a leading cause of disability and mortality worldwide, with current therapies limited in addressing its complex pathophysiological mechanisms, such as inflammation, oxidative stress, apoptosis, and impaired autophagy. Glycyrrhizic acid (GA), a bioactive compound from licorice (*Glycyrrhiza glabra* L.), has demonstrated neuroprotective properties in preclinical studies. This review consolidates current evidence on GA’s pharmacological mechanisms and assesses its potential as a therapeutic agent for ischemic stroke. **Methods**: This review examines findings from recent preclinical studies and reviews on GA’s neuroprotective effects, focusing on its modulation of inflammation, oxidative stress, apoptosis, and autophagy. Studies were identified from major scientific databases, including PubMed, Web of Science, and Embase, covering research from January 2000 to August 2024. **Results**: GA has demonstrated significant neuroprotective effects through the modulation of key pathways, including HMGB1/TLR4/NF-κB and Keap1/Nrf2, thereby reducing neuroinflammation, oxidative stress, and apoptosis. Additionally, GA promotes autophagy and modulates immune responses, suggesting it could serve as an adjunct therapy to enhance the efficacy and safety of existing treatments, such as thrombolysis. **Conclusions**: Current findings underscore GA’s potential as a multi-targeted neuroprotective agent in ischemic stroke, highlighting its anti-inflammatory, antioxidant, and anti-apoptotic properties. However, while preclinical data are promising, further clinical trials are necessary to validate GA’s therapeutic potential in humans. This review provides a comprehensive overview of GA’s mechanisms of action, proposing directions for future research to explore its role in ischemic stroke management.

## 1. Introduction

Ischemic stroke is the second leading cause of death globally and a significant contributor to long-term disability, affecting millions of individuals each year. The World Health Organization (WHO) reports that approximately 15 million people experience strokes annually, with 5 million fatalities and another 5 million left permanently disabled, placing an immense burden on public health and global economies [1]. Ischemic strokes, which result from an abrupt interruption of cerebral blood flow, account for 87% of all stroke cases [2], highlighting the need for effective treatment strategies. The subsequent reperfusion of blood following ischemia often triggers cerebral ischemia-reperfusion injury (CI/RI), leading to additional damage through oxidative stress, inflammation, and apoptosis, which exacerbate the initial brain injury [3]. Although current treatments, such as thrombolytics and mechanical thrombectomy, can be effective, they are constrained by narrow therapeutic windows and potential risks, including hemorrhagic complications [4,5]. These limitations underscore the urgent need for novel, safe, and effective therapeutic approaches.

*G*. *glabra* L., commonly known as licorice, is a perennial herb extensively used in traditional medicine, particularly across Eurasia, for its nutritional and medicinal properties [6]. The primary active compound in licorice, GA, has demonstrated significant anti-inflammatory, antioxidant, and neuroprotective properties [7]. GA’s structure, comprising two stereoisomers (18α-glycyrrhizic acid and 18β-glycyrrhizic acid) (Figure 1), contributes to its diverse biological activities, including potential therapeutic effects in neurological disorders such as ischemic stroke, dementia, Alzheimer’s disease, and Parkinson’s disease [8,9]. The neuroprotective benefits of GA are largely attributed to its ability to inhibit the high mobility group box 1 (HMGB1) protein, a key player in inflammatory responses, as well as its capacity to scavenge reactive oxygen species (ROS) and reduce neuroinflammation [10,11].

As summarized in Table 1, GA demonstrates a wide range of neuroprotective effects across various disease models, including the inhibition of inflammatory pathways and reduction of infarct size. Despite the promising potential of GA, a systematic understanding of its neuroprotective mechanisms, particularly in the context of ischemic stroke and CI/RI, remains incomplete. This review consolidates current knowledge on the pharmacological effects of GA and its active components on neurological disorders, with a particular focus on ischemic stroke and CI/RI [12,13]. By elucidating the mechanisms through which GA exerts its neuroprotective effects, this review aims to explore GA’s potential as a therapeutic agent for these debilitating conditions, offering insights into its broader applicability in managing ischemic injuries and improving patient outcomes.

## 2. Article Search Strategy Process and Study Selection

A comprehensive literature search was conducted to identify studies examining the neuroprotective effects of GA in ischemic stroke and CI/RI. The search was performed using three major databases, Web of Science, PubMed, and Embase, covering publications from 1 January 2000 to 31 August 2024. Keywords used in the search included “Glycyrrhizic acid”, “Cerebral ischemia-reperfusion”, and “Ischemic stroke”. The inclusion criteria were original research studies and reviews focusing on the effects of GA on ischemic stroke and CI/RI published in English. Studies were excluded if they were irrelevant to the topic, lacked sufficient data, or were published in languages other than English.

A total of 109 articles were initially identified through this search strategy. Following a thorough screening process based on relevance, quality, and alignment with the study’s objectives, 46 articles were selected for detailed review. These selected articles provided comprehensive insights into the potential of GA as a therapeutic agent for ischemic stroke and CI/RI, covering various aspects such as its anti-inflammatory, antioxidant, and neuroprotective properties. This systematic approach ensured that the most relevant and high-quality studies were included in the review, allowing for a detailed analysis of GA’s therapeutic potential.

The detailed search strategies for each database are provided in Appendix A, and the flow of the screening process is illustrated in Figure 2. This meticulous selection process supports this review’s objective to consolidate current knowledge on GA’s pharmacological effects and to explore its potential as a treatment for ischemic stroke and CI/RI.

## 3. Therapeutic Potential of Glycyrrhizic Acid in Ischemic Stroke

### 3.1. Glycyrrhizic Acid in Modulating Inflammation in Ischemic Stroke

HMGB1, a non-histone DNA-binding nuclear protein, plays a crucial role in the pathology of CI/RI. Recognized as a damage-associated molecular pattern (DAMP), HMGB1 is actively secreted by immune cells or passively released from necrotic cells during ischemic events. Its binding to pattern recognition receptors, such as Toll-like receptor 2/4 (TLR2/4) and the receptor for advanced glycation end-products (RAGE), triggers innate immune responses that promote the release of pro-inflammatory cytokines and exacerbate tissue damage, amplifying the inflammatory cascade in affected brain regions [35,36]. When HMGB1 binds to the RAGE, it activates several downstream signaling pathways, including JAK/STAT, ERK1/2, JNK/SAPK, and MAPK. These pathways are crucial for the activation of transcription factors such as nuclear factor kappa B (NF-κB) and STAT3, which facilitate the translocation of inflammatory cells and the production of inflammatory mediators, exacerbating systemic inflammatory responses [37].

Under ischemic conditions, HMGB1 binds to the TLR4 receptor, initiating a critical inflammatory response. This binding activates microglia, the resident immune cells of the central nervous system, which play a pivotal role in responding to ischemic damage [38,39,40]. Typically, during ischemic conditions, activated microglia adopt a pro-inflammatory (M1) state, releasing cytokines such as TNF-α and IL-6 through intracellular pathways such as MyD88 and TRIF, which in turn activate NF-κB. The activation of NF-κB, particularly its p65 subunit, further intensifies the inflammatory response, significantly contributing to neuronal damage and cell death [41,42].

In pathological conditions, such as those induced by CI/RI, activated microglia often adopt a pro-inflammatory (M1) phenotype, characterized by the release of inflammatory chemokines and cytokines, including CD16 and iNOS, which contribute significantly to neuronal damage. Conversely, M2 microglia, marked by indicators such as CD206 and Arg-1, play a crucial role in tissue repair and debris clearance, mediating anti-inflammatory effects essential for recovery [43,44].

GA, the principal bioactive component of licorice, acts as a potent inhibitor of HMGB1 by directly binding to the HMG box domains, forming a complex that effectively inhibits HMGB1’s phosphorylation, secretion, and translocation from the nucleus to the cytoplasm. This direct inhibition contrasts with HMGB1 antibodies, which primarily block HMGB1’s interaction with the RAGE receptor and thus possess a narrower scope of action [7,25,45,46]. GA’s ability to inhibit HMGB1 expression has also been shown to reduce memory deficits and alleviate neurological impairments in animal models of neuroinflammation induced by lipopolysaccharide (LPS), highlighting its potential for managing inflammation-related neurological conditions [47]. By blocking the interaction between HMGB1 and RAGE, GA prevents NF-κB activation, reduces its transcriptional activity, and subsequently lowers the production of inflammatory cytokines, ultimately suppressing the overall inflammatory response [25]. GA also modulates microglial dynamics by promoting a shift from the pro-inflammatory M1 state to the reparative M2 state, thereby reducing inflammation and enhancing recovery processes within the brain following a stroke. This GA-induced shift minimizes ischemic damage and supports long-term neurological function, underscoring GA’s potential as a neuroprotective agent (Figure 3) [11,14].

Studies indicate that enriched housing conditions after a stroke promote neurogenesis and functional recovery by inhibiting calpain 1 activity and activating the STAT3/HIF-1α/VEGF signaling pathway. GA further enhances these recovery processes by modulating HMGB1-related pathways, which are critical in neurogenesis and neuroinflammation. Through the inhibition of HMGB1, GA plays a neuroprotective role by reducing inflammatory responses and supporting cellular repair mechanisms, illustrating its therapeutic potential for improving post-stroke outcomes in ischemic stroke models [48].

HMGB1 triggers pyroptosis, a form of programmed inflammatory cell death, through inflammasome activation. This process exacerbates neuronal injury by releasing additional pro-inflammatory cytokines, thereby amplifying neuroinflammation and tissue damage in ischemic stroke [49]. Research by Xiong et al. demonstrates that GA mitigates these effects by targeting HMGB1-mediated T cell activity, significantly reducing CD4 and CD8 T cell infiltration and lowering levels of pro-inflammatory cytokines, such as IFNγ. This dual action of inhibiting pyroptosis and modulating T cell infiltration aligns with GA’s role in reducing neuroinflammation and highlights its neuroprotective potential in cerebral ischemia models through immune regulation [12].

Glycyrrhizin also plays a crucial role in modulating the HMGB1-TLR4-IL-17A signaling pathway, which mediates inflammatory responses and neuronal apoptosis during ischemic stroke. Recent studies have demonstrated that GA significantly alleviates neurotoxicity by inhibiting HMGB1 phosphorylation. This inhibition prevents HMGB1 from translocating to the cytoplasm, thereby reducing the activation of downstream inflammatory pathways, including NF-κB [50]. By inhibiting this pathway, glycyrrhizin significantly reduces IL-17A secretion, decreases infarct size, improves neurological outcomes, and reduces neuronal apoptosis, highlighting its therapeutic potential in stroke management [17,20].

Post-treatment with Glycyrrhizae Radix et Rhizoma (GRex), particularly its methanolic extract, has shown neuroprotective effects in ischemic stroke models. These include reducing cerebral infarction, alleviating cerebral edema, and modulating inflammatory responses through the regulation of microglia and astrocytes [51]. Specifically, GRex dampens inflammatory responses by inhibiting TLR4 and NF-κB signaling pathways, mechanisms closely aligned with GA’s known anti-inflammatory actions. Furthermore, GA exhibits protective effects against hemorrhagic transformation (HT), notably in models of delayed thrombolytic therapy. When administered alongside delayed tissue plasminogen activator (t-PA) treatments, GA inhibits peroxynitrite-mediated HMGB1/TLR2 signaling cascades, thereby reducing blood-brain barrier (BBB) permeability, preserving key structural proteins such as collagen IV and claudin-5, and suppressing matrix metalloproteinase-9 (MMP-9) activation. These actions decrease the likelihood of HT, lower mortality, and improve neurological recovery, highlighting GA’s potential as an effective adjuvant to thrombolytic therapy by mitigating the adverse effects commonly associated with delayed t-PA administration [15,52].

Gualou Guizhi granules (GLGZG) and Trichosanthes kirilowii cassia twig particles, both containing GA, have also shown promising neuroprotective effects in ischemia/reperfusion (I/R) injury models. GLGZG administration enhances BBB integrity, reduces infarct volume, and alleviates neurological deficits in MCAO-induced I/R injury [53]. Likewise, Trichosanthes kirilowii cassia twig particles cross the BBB, allowing active components such as paeoniflorin, albiflorin, and liquiritin to reduce neurological deficits, BBB permeability, and infarct size [54]. These findings highlight the potential of GA and its derivatives to modulate BBB permeability and deliver multifaceted neuroprotective actions in ischemic conditions, suggesting their promise in combination therapies for ischemic stroke recovery.

In cerebral ischemia models, particularly in diabetic mice, post-treatment with GA has been found to significantly suppress the expression of key components in the HMGB1/TLR4-Myd88/NF-κB p65 signaling pathway. This suppression effectively reduces ischemic brain damage and improves neurological outcomes [16]. Additionally, recent research has underscored the critical role of the HMGB1–TLR2/4 axis in microglial activation following cortical spreading depression—a phenomenon that exacerbates neuronal injury in conditions such as stroke. Glycyrrhizin’s inhibition of this pathway has been shown to prevent microglial hypertrophy and mitigate associated neuroinflammatory damage, further highlighting its potential as a neuroprotective agent in ischemic conditions [27].

In studies of neonatal HIBD, GA has been shown to inhibit additional inflammatory pathways, such as the HMGB1/GPX4 pathway. By inhibiting HMGB1 expression and ectopia, GA blocks the p38 MAPK signaling pathway, which is involved in cell proliferation, differentiation, apoptosis, necrosis, and cell cycle regulation. These actions suggest that GA can effectively mitigate neuronal damage and provide neuroprotection in conditions characterized by heightened inflammation [21,28].

HMGB1 also plays a role in neurovascular repair by influencing the paracrine functions of endothelial progenitor cells (EPCs) after ischemic stroke [55]. HMGB1 released from activated microglia can stimulate IL-8 production in EPCs, thereby enhancing their angiogenic potential and supporting neurovascular remodeling. By inhibiting HMGB1, GA may attenuate this EPC-driven angiogenesis, suggesting that GA’s therapeutic scope extends beyond inflammation control to modulation of neurovascular repair mechanisms, enriching its neuroprotective profile in stroke management.

While primarily known for its role in neuroinflammation, HMGB1 also functions as a critical autocrine trophic factor for oligodendrocytes (OLs) under ischemic stress, especially in white matter stroke models [56]. HMGB1 released by damaged OLs activates TLR2 signaling in nearby OLs, promoting cell survival and supporting myelin integrity. In this context, GA’s inhibition of HMGB1 may inadvertently interfere with this protective mechanism in white matter, potentially exacerbating demyelination and related functional deficits under ischemic conditions. This complexity suggests that although GA’s inhibition of HMGB1 provides significant anti-inflammatory benefits in gray matter regions, its effects on white matter injury might be more nuanced, as it could disrupt OL-specific autocrine support essential for myelin preservation and structural stability during ischemic stress.

In addition to its effects in ischemic stroke, glycyrrhizin has shown promising results in models of traumatic brain injury (TBI). Glycyrrhizin’s inhibition of HMGB1 reduces brain edema, astrocyte and microglia activation, and cognitive deficits in models of pediatric TBI, suggesting that glycyrrhizin can provide long-term neuroprotection by mitigating neuroinflammatory responses [57]. Furthermore, studies on TBI models provide additional evidence of GA’s neuroprotective effects via modulation of the HMGB1/TLR4/RAGE/NF-κB pathway. Research by Gu et al. demonstrated that GA administration in TBI significantly reduced the expression of HMGB1 and its receptors TLR4 and RAGE, thereby downregulating NF-κB activity and reducing levels of pro-inflammatory cytokines, such as TNF-α, IL-1β, and IL-6 [58]. These results highlight GA’s potential in mitigating secondary brain injury by modulating inflammation and apoptosis, reinforcing its therapeutic applicability in ischemic and traumatic neural injuries. This aligns with findings that glycyrrhizin effectively reduces HMGB1–RAGE interactions, leading to decreased expression of inflammatory cytokines such as TNF-α, IL-1β, and IL-6, which are critical mediators of inflammation in TBI [25].

Beyond cerebral ischemia, glycyrrhizin’s efficacy extends to other injury models, such as spinal cord injuries and thermal burns. In spinal cord injury models, GA treatment significantly decreases pro-inflammatory markers such as iNOS and CD86 while simultaneously increasing anti-inflammatory markers such as CD206 and Arg-1 within microglia [59,60]. These findings suggest that GA effectively inhibits the polarization of microglia towards the pro-inflammatory M1 phenotype and promotes their shift toward the reparative M2 phenotype. This modulation is crucial in treating spinal cord injuries and offers a new direction for using GA in managing CI/RI by leveraging its broad anti-inflammatory and neuroprotective capabilities.

In models of thermal burns, glycyrrhizin suppresses systemic inflammation by inhibiting HMGB1 and reducing pro-inflammatory cytokine levels, thereby protecting vital organs from damage [61]. Additionally, glycyrrhizin inhibits T cell proliferation and reduces markers such as CD68 and MPO, further controlling inflammation at the molecular level by preventing inflammasome activation [12].

In myocardial ischemia-reperfusion injury models, glycyrrhizin blocks the HMGB1-dependent phospho-JNK/Bax pathway, reducing apoptosis and highlighting its potential across ischemic conditions [30]. In hyperglycemic and diabetic stroke models, glycyrrhizin alleviates ischemic damage by preserving the blood-brain barrier and reducing neuroinflammation, making it a promising therapeutic agent for stroke patients with complicating conditions such as diabetes [62,63].

Diammonium glycyrrhizinate (DG), a derivative of GA, has also demonstrated significant neuroprotective effects. In a mouse model of ischemia-reperfusion injury, DG was shown to reduce the expression of inflammatory mediators such as IL-1, TNF-α, COX-2, iNOS, and NF-κB, leading to improved neurological outcomes and reduced brain injury [64]. Recent studies further indicate that DG exhibits neurovascular protective effects in cerebral I/R models by upregulating tight junction proteins such as Rac-1, Claudin-5, and VE-Cadherin, thereby reducing BBB permeability and supporting neurovascular stability [65]. This dual action, which includes both the suppression of inflammatory mediators and enhancement of BBB integrity, aligns with GA’s known effects on neuroinflammation and suggests that GA and its derivatives may be valuable therapeutic agents for managing ischemic injuries and supporting neurovascular stability.

GA has demonstrated significant neuroprotective effects across various models of neurological injury by inhibiting the release and activity of HMGB1. The role of HMGB1 in ischemic stroke has been extensively documented, with its release contributing to neuroinflammation and blood-brain barrier disruption—processes that glycyrrhizin effectively mitigates. In focal cerebral ischemia, GA reduces brain infarction by inhibiting T cell functions and preventing the release of HMGB1, thus dampening subsequent inflammatory responses [66]. This is further supported by evidence highlighting HMGB1’s dual role in the early inflammatory response and neurovascular remodeling during stroke recovery, underscoring the potential of targeting HMGB1 as a therapeutic strategy to reduce stroke-induced damage [67,68].

### 3.2. Glycyrrhizic Acid in Modulating Apoptosis in Ischemic Stroke

Apoptosis, or programmed cell death, is essential for maintaining cellular homeostasis. However, during ischemic stroke, excessive apoptosis exacerbates neuronal loss and contributes to further brain damage. This process is distinct from necrosis, which typically induces inflammation. In apoptosis, cells undergo controlled degradation involving caspase activation, nuclear condensation, and DNA fragmentation [69]. In contrast, necrosis triggers the release of DAMPs, such as HMGB1, which activate inflammatory responses [70].

During ischemic stroke, neurons release HMGB1, a key DAMP that exacerbates inflammation and promotes apoptosis. HMGB1 activates microglia, leading to additional inflammatory responses that damage surrounding neurons [71,72]. Targeting HMGB1 signaling, specifically through the TLR4 receptor, has been shown to reduce neuronal death. GA, by inhibiting HMGB1, can reduce neuronal apoptosis in ischemic conditions by modulating the HMGB1/TLR4 axis [18,73].

DG, a derivative of GA, reduces apoptosis by modulating key signaling pathways such as Akt and caspase-3, which are crucial for cell survival and programmed cell death during ischemic events [74]. Targeting these pathways is relevant not only for stroke but also for neurodegenerative diseases such as Alzheimer’s disease [75]. Furthermore, HMGB1 released during ischemia impairs astrocytic glutamate clearance, leading to excitotoxicity. GA mitigates this process by inhibiting the HMGB1/TLR4 axis [76].

In ischemic stroke, the balance between pro-apoptotic and anti-apoptotic proteins, Bax and Bcl2, respectively, is critical. A higher Bax/Bcl2 ratio promotes apoptosis, while an increased Bcl2/Bax ratio has been shown to protect neurons and reduce infarct size [77,78]. GA effectively modulates this balance by increasing Bcl2 expression and reducing Bax expression, thereby inhibiting apoptosis [19,79]. Studies on neonatal mice subjected to ischemia/hypoxia confirm that GA regulates Bax and Bcl2 protein levels, preventing neuronal apoptosis in the hippocampus [23].

Mitochondrial dysfunction is a key factor in ischemic stroke-induced apoptosis. The release of cytochrome c from mitochondria into the cytoplasm triggers caspase-3 activation, leading to apoptosis [80]. GA inhibits cytochrome c release, preventing downstream activation of apoptotic pathways [26]. In ischemia-reperfusion injury models, GA also inhibits the JNK and p38-MAPK pathways, reducing oxidative stress and inflammation and ultimately decreasing apoptosis [22].

These findings are consistent with other research where roasted licorice ethanol extract (rLee) demonstrated significant neuroprotective effects in a gerbil model of transient forebrain ischemia, preserving neuronal density in the hippocampal CA1 region by reducing ischemic damage [81]. Additionally, DG has been shown to improve survival outcomes in ischemic models, such as random skin flaps in rats, through mechanisms that reduce inflammation and oxidative stress and promote angiogenesis [82].

In spinal cord injuries, inhibiting HMGB1 with GA has been shown to reduce edema and astrocyte activation, providing protection against secondary injury [79]. Similarly, GA reduces apoptosis in ischemic stroke by preventing HMGB1-induced excitotoxicity and inflammation [22]. This ability to inhibit HMGB1’s pro-apoptotic effects extends beyond stroke, as demonstrated in TBI models, where GA mitigates brain edema and reduces the activation of microglia and astrocytes [57].

GA has also shown efficacy in chronic cerebral ischemia models by improving cognitive function and reducing apoptosis. GA achieves these effects by regulating the PI3K/Akt/GSK3β pathway, reducing oxidative stress, and inhibiting cytochrome c release [26]. Furthermore, in neonatal brain injury models, GA prevents neuronal loss and preserves myelin integrity, which is essential for proper brain function [23].

While GA primarily reduces apoptosis through caspase-dependent pathways, it also modulates cell death through non-caspase-dependent mechanisms [83,84]. For example, GA has demonstrated anti-proliferative and pro-apoptotic effects in non-neurological conditions, such as cancer, suggesting broader therapeutic applications beyond ischemic stroke [85].

### 3.3. Glycyrrhizic Acid in Modulating Oxidative Stress and Autophagy in Ischemic Stroke

GA shows significant potential in modulating oxidative stress and autophagy, both critical factors in ischemic stroke. Studies indicate that licorice treatment in ischemic models significantly increases superoxide dismutase (SOD) activity and reduces lactate dehydrogenase (LDH) release, supporting the antioxidant properties of GA and its derivatives [26,81]. Research by Namik et al. on 18β-glycyrrhetinic acid (18β-GA), a GA metabolite, highlighted its protective effects against oxidative damage induced by global cerebral I/R injury. This study demonstrated that 18β-GA significantly attenuates lipid peroxidation, increases SOD, catalase (CAT), and glutathione (GSH) activity, and reduces neuronal apoptosis in I/R-injured brain tissue, underscoring the neuroprotective potential of GA and its derivatives in managing oxidative stress [86]. Additionally, carbenoxolone (CBX) [58], a semisynthetic GA derivative, exhibits protective effects against I/R injury by reducing oxidative damage. CBX decreases malondialdehyde (MDA) levels, a marker of lipid peroxidation, in both hippocampal and skeletal muscle tissues during ischemic events, mitigating cellular damage. These antioxidant properties highlight the therapeutic scope of GA derivatives in reducing oxidative stress and promoting neuroprotection in ischemic stroke models.

ROS exacerbate ischemic injury by promoting cellular apoptosis, calcium overload, and pro-inflammatory cytokine secretion. In neonatal HIBD, GA regulates HMGB1 expression through microglial polarization, inhibiting ROS generation and reducing neuronal damage [14]. GA’s inhibitory effects on HMGB1 also attenuate spinal cord edema and astrocyte activation in injury models [87]. Furthermore, GA demonstrates antioxidant properties in models of neurodegenerative diseases, such as Huntington’s disease, by downregulating the HMGB1/TLR4/NF-κB pathway, reducing markers of oxidative stress such as MDA, and increasing antioxidant defenses such as GSH and SOD, underscoring GA’s broad neuroprotective potential [29].

Oxidative stress, driven by ROS, plays a central role in ischemia-reperfusion injury, where elevated ROS production leads to tissue damage, apoptosis, calcium overload, and pro-inflammatory cytokine secretion, all of which exacerbate ischemic injury. The NRF2 (Nuclear factor erythroid 2–related factor 2) and KEAP1 (Kelch-like ECH-associated protein 1) signaling pathway is essential in cellular responses to oxidative stress by regulating the production of antioxidants and detoxification enzymes. Under normal conditions, NRF2 is bound to KEAP1 in the cytoplasm, where KEAP1 promotes NRF2 degradation. However, during oxidative stress, KEAP1 releases NRF2, enabling it to translocate to the nucleus. In the nucleus, NRF2 activates antioxidant response elements (AREs), which enhance the expression of cytoprotective enzymes such as heme oxygenase-1 (HO-1), SOD, and CAT. These enzymes collectively reduce ROS and mitigate cellular damage [35,88]. GA’s modulation of the KEAP1/NRF2 pathway enhances its antioxidant capacity, positioning it as a neuroprotective agent in ischemic conditions by reducing ROS levels, protecting mitochondrial integrity, and improving cellular resilience. However, persistent NRF2 activation can lead to nuclear accumulation and binding to the Klf9 promoter, which can inadvertently increase ROS and lead to cell death under certain pathological conditions [31,32,89,90]. This dual role of NRF2 highlights the complexity of its regulation, particularly as it pertains to neurodegenerative diseases such as multiple sclerosis, Parkinson’s disease, Alzheimer’s disease, and amyotrophic lateral sclerosis (ALS) [91,92].

Beyond antioxidant protection, GA’s influence on the KEAP1/NRF2 pathway aligns with its broader effects on mitochondrial health through pathways such as PINK1/Parkin-dependent mitophagy, a selective autophagic process critical for removing damaged mitochondria. Autophagy is a crucial cellular mechanism for maintaining homeostasis by degrading and recycling damaged organelles and proteins, especially under ischemic conditions. In ischemic stroke, mitochondrial damage results in increased ROS production, which in turn promotes further cell death. Through the PINK1/Parkin-dependent pathway, GA promotes mitophagy, removing dysfunctional mitochondria and shielding cells from oxidative damage [93]. Under ischemic conditions, PINK1 accumulation on the outer mitochondrial membrane signals the initiation of mitophagy, which not only reduces ROS production but also prevents cellular apoptosis [94]. GA’s activation of this pathway has demonstrated neuroprotective effects during cerebral ischemia-reperfusion injury and in neurodegenerative diseases such as Parkinson’s disease, where it regulates PINK1 and parkin expression levels, further supporting neuronal survival [95].

Additionally, GA’s regulatory impact extends to interactions between HMGB1 and Beclin-1, a critical autophagy regulator, enhancing protective autophagy while limiting excessive cell death. Endogenous HMGB1 functions as a pro-autophagic protein by binding to Beclin-1, thereby maintaining autophagic activity crucial for cell survival. GA’s ability to inhibit the MEK/ERK signaling pathway through its interaction with HMGB1 also helps prevent excessive autophagy-induced cell death, which is particularly relevant in neuroinflammatory and neurodegenerative contexts [33,34,35,76,96]. This nuanced regulation of autophagy underscores GA’s potential in supporting neuronal survival, highlighting its broader therapeutic application in both ischemic and neurodegenerative diseases.

Ferroptosis, a form of iron-dependent lipid peroxidation-induced cell death, plays a crucial role in ischemic brain injury. GA effectively inhibits ferroptosis via the HMGB1/GPX4 pathway, especially in neonatal hypoxic-ischemic brain injury. GPX4, an enzyme that reduces lipid peroxides using GSH, is vital in protecting cells from oxidative stress-induced damage. By enhancing GPX4 activity, GA mitigates lipid peroxidation accumulation and preserves mitochondrial integrity, thereby reducing neuronal injury [21]. GA’s inhibition of ferroptosis is particularly important, as ferroptotic cell death is associated with mitochondrial dysfunction, ROS accumulation, and iron dysregulation, all of which are exacerbated during ischemic events. GA also enhances antioxidant enzyme activity, further reducing oxidative damage and preventing ferroptosis-induced neurodegeneration [21,29]. These studies underscore GA’s protective effects in ischemic brain injuries, extending beyond its antioxidant properties to include inhibition of ferroptosis, a critical factor in mitigating neuronal death and promoting recovery in ischemic conditions (Figure 4).

In addition to its effects in stroke models, GA has been shown to activate the Sirtuin3 (Sirt3) pathway in juvenile epilepsy models, promoting mitophagy and reducing oxidative stress. This regulation of mitochondrial autophagy not only mitigates neuronal damage but also preserves hippocampal function [97]. Moreover, the metabolite of glycyrrhizin, 18β-GA, has demonstrated the ability to inhibit apoptosis and activate protective autophagy through suppression of the JAK2/STAT3 pathway in cerebral ischemia-reperfusion injury models [13]. These findings underscore the broader therapeutic potential of GA, not only in stroke but also in managing other neuroinflammatory and neurodegenerative diseases through its dual regulation of autophagy and oxidative stress.

The clinical relevance of GA has also been validated in trials. Ravanfar et al. conducted randomized, double-blind, placebo-controlled trials demonstrating that the oral administration of licorice extract significantly improved neurological outcomes in patients with acute ischemic stroke. Patients treated with licorice extract exhibited greater reductions in NIHSS (National Institutes of Health Stroke Scale) and modified Rankin scale (MRS) scores compared to the control group, providing robust clinical evidence for GA’s neuroprotective effects. These findings underscore GA’s efficacy in reducing inflammation and oxidative stress in clinical settings, further supporting its potential as a therapeutic agent for stroke recovery [98].

A study also demonstrated that DG, a derivative of GA, significantly reduced oxidative stress and prevented histological damage in brain tissue during global ischemia-reperfusion injury, further bolstering GA’s therapeutic prospects in ischemic stroke treatment [74]. GA’s ability to reduce HT following ischemic stroke has also been explored. HT, a severe complication of thrombolysis therapy using t-PA, can be mitigated by targeting key inflammatory pathways such as HMGB1, TLR4, and NF-κB. Glycyrrhizin, by downregulating these pathways, has been shown to reduce the risk of HT and improve neurological outcomes in ischemic stroke patients [99].

GA’s neuroprotective potential extends to preventing excitotoxicity and neuronal damage by mitigating ischemia/reperfusion-induced impairment of astrocytic glutamate clearance, which is regulated through the HMGB1/TLR4 signaling axis [76]. These findings highlight the broad therapeutic potential of GA, not only in stroke treatment but also in managing other neuroinflammatory and neurodegenerative diseases.

## 4. Discussion

Ischemic stroke remains a leading cause of death and long-term disability worldwide, significantly impacting public health and the global economy [1]. Despite advances in thrombolytic therapies such as t-PA, the narrow therapeutic window and high risk of HT limit their utility [4,5]. Consequently, there is an urgent need for novel therapeutic agents that can extend the treatment window, reduce inflammation, and protect against ischemia-reperfusion injury (IRI). GA, a bioactive compound derived from licorice (*Glycyrrhiza glabra* L.), has garnered attention due to its potent neuroprotective properties, including anti-inflammatory, antioxidant, and apoptosis-regulating effects [100]. GA’s versatility, particularly in modulating multiple pathological pathways, makes it a promising candidate for stroke therapy.

GA’s ability to inhibit HMGB1 protein, a key mediator of inflammation, underscores its potential in treating ischemic stroke [10,12]. HMGB1 is released during ischemic events and amplifies neuroinflammation through interactions with TLRs and the RAGE [18,25]. By inhibiting HMGB1 and its downstream signaling via the TLR4 and MyD88 pathways, GA suppresses the release of pro-inflammatory cytokines such as TNF-α, IL-1β, and IL-6, thereby reducing inflammation [41,42]. Moreover, GA modulates the microglial response, shifting it from a pro-inflammatory (M1) to an anti-inflammatory (M2) phenotype, which is essential for reducing neuronal damage and supporting tissue repair during stroke recovery [11,14].

In addition to its anti-inflammatory effects, GA has significant antioxidant properties, primarily through activation of the Keap1/Nrf2 pathway. This pathway upregulates antioxidant enzymes such as SOD and HO-1, which are critical for neutralizing ROS generated during ischemia-reperfusion injury [31,32]. By reducing ROS levels, GA helps protect mitochondrial function, a crucial factor in preventing further cellular damage [91,92]. Mitochondrial dysfunction is a hallmark of ischemic stroke, and GA’s ability to promote mitochondrial health preserves neuronal function and mitigates oxidative stress, which exacerbates ischemic damage [21].

Apoptosis is another key pathological process in ischemic stroke, contributing to extensive neuronal loss [69]. GA modulates apoptotic pathways by balancing pro-apoptotic and anti-apoptotic proteins, particularly Bax and Bcl-2 [19,79]. By inhibiting cytochrome c release from mitochondria and blocking caspase-3 activation, GA prevents the initiation of apoptosis, thereby reducing cell death [26]. This anti-apoptotic effect is particularly beneficial in the context of delayed reperfusion therapies, where prolonged ischemia increases the risk of apoptosis.

Furthermore, GA’s role in promoting autophagy adds to its neuroprotective effects [93]. Autophagy is a cellular process that degrades and recycles damaged cellular components, helping to maintain cellular homeostasis under stress conditions such as ischemia. GA has been shown to enhance mitophagy—the selective removal of damaged mitochondria—via the PINK1/Parkin pathway, reducing oxidative stress and preventing further neuronal injury [93,95]. This dual action of GA in regulating apoptosis and promoting autophagy underscores its comprehensive neuroprotective potential in ischemic conditions.

Beyond its applications in ischemic stroke, GA shows broader therapeutic potential in treating other ischemic and inflammatory conditions. Its ability to inhibit HMGB1 signaling and regulate key inflammatory and oxidative pathways suggests that GA could be useful in treating diseases characterized by excessive inflammation and oxidative stress, such as neurodegenerative diseases and cardiovascular disorders [101,102]. Additionally, GA’s anti-proliferative and pro-apoptotic effects observed in cancer models indicate potential utility beyond neurological conditions, opening avenues for its application in oncology [33,85,103].

## 5. Conclusions

In conclusion, GA presents as a promising therapeutic agent for ischemic stroke due to its ability to modulate multiple pathological processes, including inflammation, oxidative stress, apoptosis, and autophagy. GA’s potential to cross the blood-brain barrier and its cost-effective synthesis enhance its clinical applicability. Preclinical studies have demonstrated GA’s efficacy in reducing infarct size, alleviating neuroinflammation, and improving functional outcomes. However, large-scale clinical trials are required to confirm these findings in human patients. Furthermore, exploring GA as an adjunct therapy to current treatments, such as t-PA, could reduce complications such as HT and improve recovery outcomes. GA’s broader applications in treating neurodegenerative diseases and cancers make it a versatile candidate for future therapeutic development.

## Figures and Tables

**Figure 1 pharmaceuticals-17-01493-f001:**
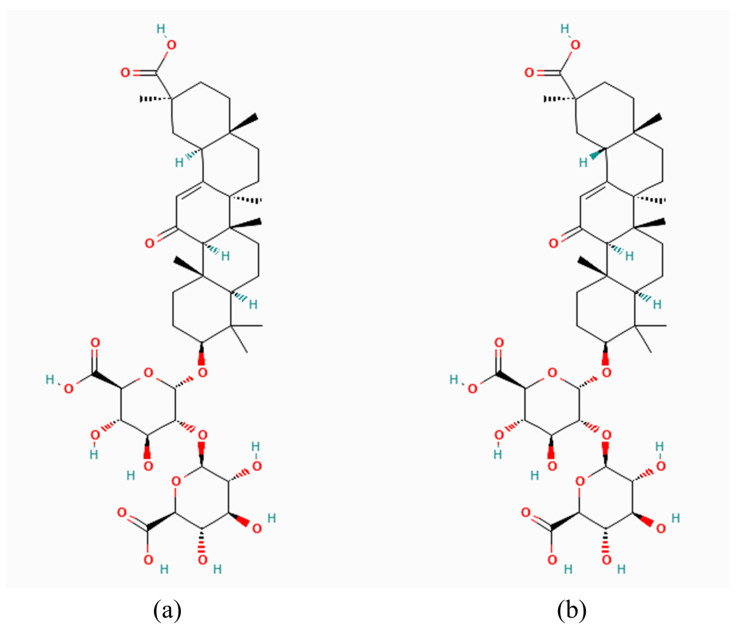
Structural configurations of glycyrrhizic acid stereoisomers. (**a**) 18α-Glycyrrhizic Acid: The α stereoisomer of glycyrrhizic acid, featuring a specific spatial arrangement around the 18th carbon atom. (**b**) 18β-Glycyrrhizic Acid: The β stereoisomer, which is the predominant and biologically active form found in licorice root. Glycyrrhizic acid generally refers to this 18β form, known for its anti-inflammatory, antiviral, and hepatoprotective properties. Structural diagrams were downloaded from PubChem.

**Figure 2 pharmaceuticals-17-01493-f002:**
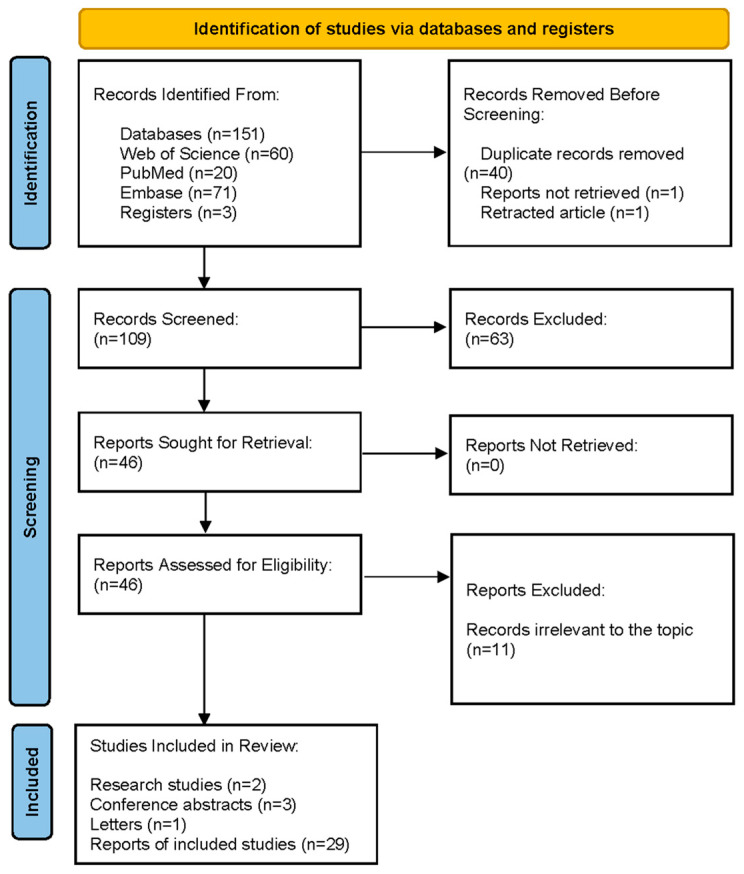
PRISMA flow diagram illustrating the study selection process.

**Figure 3 pharmaceuticals-17-01493-f003:**
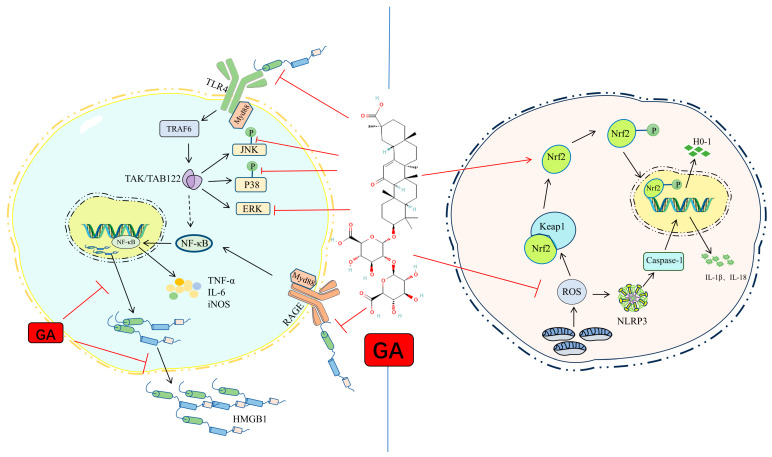
Schematic illustration showing the anti-inflammatory and antioxidant effects of GA. On the left, GA inhibits the binding of HMGB1 to TLR4 and RAGE receptors, blocking downstream signaling cascades such as MyD88/TRAF6 and NF-κB activation, thereby reducing pro-inflammatory cytokine production. On the right, GA activates the Keap1/Nrf2 pathway, leading to increased expression of antioxidant enzymes (e.g., HO-1 and SOD), which reduce ROS and mitigate oxidative stress. Red T-shapes represent the inhibition of molecule expression, red and black arrows represent the promotion of molecule expression, and dashed black arrows indicate molecules with potential promotive influence.

**Figure 4 pharmaceuticals-17-01493-f004:**
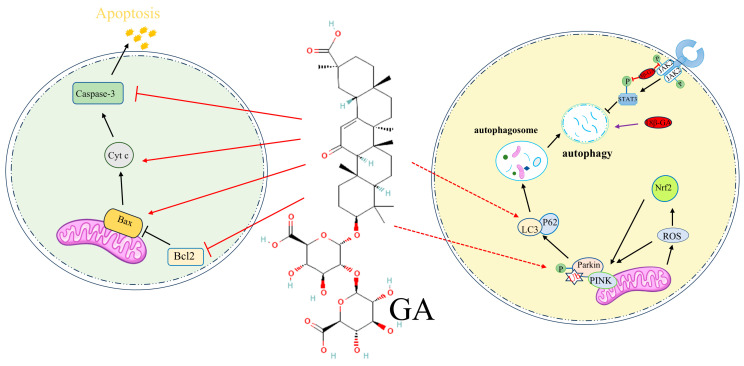
Schematic representation of GA’s role in regulating apoptosis and autophagy. On the left, GA inhibits mitochondrial cytochrome c release and caspase-3 activation by modulating the Bax/Bcl-2 ratio, thus reducing neuronal apoptosis. On the right, GA promotes autophagy via the Keap1/Nrf2 and PINK1/Parkin pathway, facilitating the clearance of damaged mitochondria and reducing oxidative stress, contributing to enhanced neuronal survival. Red and black T-shapes represent the inhibition of molecule expression, red and black arrows indicate the promotion of molecule expression, and dotted red arrows represent molecules that may promote additional effects.

**Table 1 pharmaceuticals-17-01493-t001:** Neuroprotective and Therapeutic Effects of GA Across Various Disease Models.

Disease/Condition	Experimental Material	Effect	Reference
hypoxic-ischemic brain damage (HIBD)	HAPI cells,	GA inhibits M1 microglial polarization while promoting M2 microglial polarization, suppresses ROS production, reduces neuronal damage, and significantly alleviates cerebral edema and infarction (Infarct size was reduced by an estimated 7%).	[14]
	Sprague-Dawley (SD) rats		
Ischemic stroke	Male Sprague-Dawley rats	GA inhibits the ONOO−/HMGB1/TLR2 signaling cascade to reduce hemorrhagic transformation and improve neurological function.	[15]
Cerebral ischemia-reperfusion injury	Male C57BL/6J mice	GA significantly inhibits the HMGB1/TLR4-Myd88/NF-κB p65 signaling pathway, reducing ischemic brain injury and improving neurological outcomes in diabetic mice. The infarct volume in the cerebral cortex after ischemia was reduced by an estimated 27%.	[16]
Cerebral ischemia-reperfusion injury	Male C57BL/6J mice	GA regulates the HMGB1-TLR4-IL-17A signaling pathway, significantly decreases IL-17A secretion, reduces infarct size, improves neurological outcomes, and lowers neuronal apoptosis.	[17]
Cerebral ischemia-reperfusion injury	Male adult Sprague-Dawley rats	GA inhibits the HMGB1-mediated TLR4/NF-κB pathway, alleviating inflammation and neurological deficits. The infarct volume in the cerebral cortex after ischemia was reduced by an estimated 11%.	[18]
Cerebral ischemia-reperfusion injury	PC12 cells, Male Wistar rats	GA metabolite 18β-glycyrrhetinic acid (18β-GA) provides protection in ischemia-reperfusion models by inhibiting JAK2/STAT3 pathway-induced autophagy. The maximum infarct size reduction was an estimated 21%.	[13]
Ischemic injury	Rat pheochromocytoma cells (PC12)	GA reduces the mitochondrial Bax/Bcl-2 ratio and activates PI3K/Akt signaling, modulates the intracellular antioxidant system, and induces apoptosis to protect against ischemic injury.	[19]
Acute ischemic stroke	Male Wistar rats	GA inhibits HMGB1 expression and pro-inflammatory cytokines. The infarct volume in the cerebral cortex after ischemia was reduced by 69.9%.	[20]
HIBD	Male and female neonatal Sprague-Dawley rats	GA inhibits ferroptosis through the HMGB1/GPX4 pathway, reducing neuronal loss in neonatal hypoxic-ischemic brain injury.	[21]
Focal cerebral ischemia/reperfusion	Male Sprague-Dawley rats	GA inhibits JNK and p38-MAPK pathways, reducing oxidative stress, inflammation, and apoptosis in ischemia-reperfusion models. The infarct volume in the cerebral cortex after ischemia was reduced by 58% (estimated).	[22]
Focal cerebral ischemia	SD rats	GA inhibits T cell proliferation, decreases markers such as CD68 and MPO, and prevents inflammasome activation. The size of the infarcts decreased by 15% (estimated).	[12]
Hypoxia-Ischemia	ICR mice	GA modulates Bax and Bcl2 protein levels, preventing apoptosis of hippocampal neurons in neonatal ischemia/hypoxia models.	[23]
Ischemic spinal cord injury	Male Sprague-Dawley rats	GA reduces spinal cord ischemic injury by lowering inflammatory cytokines and suppressing HMGB1 release.	[24]
Traumatic brain injury	Adult male Wistar rats	GA reduces the production of inflammatory factors by inhibiting the interaction between HMGB1 and RAGE, ultimately suppressing the overall inflammatory response.	[25]
Chronic cerebral hypoperfusion (CCH)	Adult male Sprague-Dawley rats	GA reduces cytochrome c release, prevents apoptosis, and reduces oxidative stress via the GSK3β/Nrf2 signaling pathway.	[26]
Cortical spreading depression (CSD)	Male C57BL/6J mice	GA inhibits the HMGB1-TLR2/4 pathway to prevent microglial hypertrophy, reducing neuroinflammatory damage.	[27]
Epilepsy	Adult male Sprague-Dawley rats	GA exerts neuroprotective effects through the HMGB1/P38 MAPK signaling pathway.	[28]
Huntington’s disease (HD)	Adult male Wistar albino rats	GA downregulates the HMGB1/TLR4/NF-κB signaling pathway in neurodegenerative disease models, reducing oxidative stress markers such as MDA and increasing antioxidant defenses (GSH, SOD).	[29]
Myocardial ischemia-reperfusion	Male Sprague-Dawley rats	GA blocks the HMGB1-dependent P-JNK/Bax pathway, reducing apoptosis and providing myocardial protection in ischemia-reperfusion injury.	[30]
Myocardial ischemia-reperfusion	Adult male Sprague-Dawley rats	GA provides cardioprotection by activating Keap/Nrf2/HO-1 and inhibiting NF-κB signaling pathways.	[31,32]
Gastric cancer	BGC-823, SGC-7901, GES-1 cell lines	GA inhibits the proliferation of gastric cancer cells, induces apoptosis, arrests the cell cycle, and reduces colony formation by inhibiting the MEK/ERK pathway through HMGB1 interaction.	[33,34]

## Data Availability

No new data were created or analyzed in this study. Data sharing is not applicable to this article.

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
