# Peer review of "Neuroprotective Potential of Glycyrrhizic Acid in Ischemic Stroke: Mechanisms and Therapeutic Prospects"

_pharmaceuticals, 2024, doi:10.3390/ph17111493_

Round 1

Reviewer 1 Report

Comments and Suggestions for Authors

The topic is of high interest for the scientific community and is well-structured. It is pleasant to read and complete of information useful to understand the topic even if not familiar with it. I have a few suggestions for the Authors:

1) Please carefully proofread the whole manuscript for English and typos.

2) The construction of the abstract is not really appropriate for a review article. Please modify basing on previous published reviews.

3) Figure 2 shows some lines that are relative to the draft process, please remove it and replace the figure.

Comments on the Quality of English Language

There are a few English grammar errors and typos.

Author Response

Comments 1: Please carefully proofread the whole manuscript for English and typos.

Response 1: Thank you for your suggestion to carefully proofread the manuscript. We have conducted a comprehensive review and proofreading of the entire text to address any grammatical issues, typos, and inconsistencies. We ensured that technical terms, abbreviations, and references were used consistently and clarified any complex sentences for readability. Additionally, we have refined the language to improve clarity and maintain a professional tone throughout the manuscript. We hope these revisions meet your expectations for accuracy and readability.

Comments 2: The construction of the abstract is not really appropriate for a review article. Please modify basing on previous published reviews.

Response 2: Thank you for your valuable feedback on the abstract structure. We appreciate your suggestion to revise the abstract to better align with the style typically expected in review articles. Following your guidance and in adherence to the journal’s specific requirements, we have restructured the abstract to incorporate the designated headings: Background/Objectives, Methods, Results, and Conclusions. The revised abstract succinctly summarizes current knowledge on glycyrrhizic acid’s neuroprotective mechanisms and its therapeutic potential for ischemic stroke, focusing on its anti-inflammatory, antioxidant, anti-apoptotic, and autophagy-modulating properties, while avoiding detailed methodological descriptions. We ensured that the abstract objectively represents findings substantiated in the main text and avoids any overstatement of conclusions, in line with the journal’s guidelines.

Comments 3: Figure 2 shows some lines that are relative to the draft process, please remove it and replace the figure.

Response 3: Thank you for your feedback on Figure 2. I have carefully reviewed and revised the figure as per your suggestion. The revised version now excludes the draft lines that were mistakenly included in the initial submission, ensuring clarity and relevance to the final content. The updated figure aligns with the manuscript’s focus and accurately illustrates the intended concepts.

Reviewer 2 Report

Comments and Suggestions for Authors

In the current review manuscript, Liu and Sun et al. described the therapeutic effects of glycyrrhizic acid in ischemic stroke.

Ischemic stroke is a widespread disease worldwide, affecting millions of people annually, leading to death or permanent disability. For this reason, the search for new methods of preventing this disease, as well as measures supporting the treatment of its effects, is an extremely important and current problem. In relation to the mentioned problem, the authors of this manuscript focused on the therapeutic role of glycyrrhizic acid in ischemic stroke, as well as the mechanisms of its action in this area. The preparation of the review article manuscript was preceded by a thorough review of the available literature and an appropriate selection of cited English-language items. The structure of the review article raises no objections, as does the selection of cited literature, as well as the linguistic and editorial side of the study. Therefore, in light of the subject matter of the article as well as the quality of its preparation, I recommend this manuscript for publication in Pharmaceuticals as a review article.

Author Response

Thank you very much for your positive feedback and for recommending our manuscript for publication. We are grateful for your thorough evaluation of our work and are pleased that you found the manuscript well-structured and our literature review comprehensive. Your acknowledgment of the importance of exploring glycyrrhizic acid’s therapeutic potential in ischemic stroke encourages us greatly, and we are glad that the manuscript aligns well with the objectives of Pharmaceuticals.

Reviewer 3 Report

Comments and Suggestions for Authors

The work is an interesting review of the glycyrrhizic acid and Ischemic strok.

Please explain why you did not use PubMed NBCI.

Author Response

Comments 1: Please explain why you did not use PubMed NBCI.

Response 1: Thank you for your feedback and for your interest in our work. We would like to clarify that PubMed was indeed utilized as one of the primary databases in our literature search for this review. As outlined in the manuscript, we conducted a comprehensive search through PubMed alongside other major scientific databases to ensure a thorough and balanced review of glycyrrhizic acid’s therapeutic potential in ischemic stroke.

Additionally, we have detailed our specific search strategy for PubMed in Appendix A, providing a transparent overview of the keywords and inclusion criteria used in this review.

Reviewer 4 Report

Comments and Suggestions for Authors

In this manuscript authors were aimed to consolidate current knowledge on GA’s pharmacological effects in ischemic stroke, focusing on its anti-inflammatory, antioxidant, and anti-apoptotic mechanisms.

The manuscript is interesting and generally well written, and figures are clear. However, it presents some points that must be improved. In particular:

Figure 1: Did the authors draw the images or downloaded from some database?

Figure 2: it seems that this figure is a screenshot of the original image since corrections are reported and underlined. Export the image properly. 

Lines 323-329: Although NRF2/KEAP1 signaling plays a key role is the studies discussed in this review, this signaling is poorly introduced. In fact, this pathway plays a key role in the progression and onset of several diseases (see PMID: 37296999, PMID: 39034715 ).

A table summarizing the main results of the studies selected in this review should be reported

An accurate revision of typing errors is recommended

Author Response

Comments 1: Figure 1: Did the authors draw the images or downloaded from some database?

Response 1: Thank you for your question regarding Figure 1. The images in Figure 1 were sourced from the PubChem database to accurately represent the chemical structures of glycyrrhizic acid and its derivatives. We have credited the source accordingly in the manuscript to ensure proper attribution.

Comments 2: Figure 2: it seems that this figure is a screenshot of the original image since corrections are reported and underlined. Export the image properly.

Response 2: Thank you for your observation regarding Figure 2. We apologize for the oversight. The previous version of Figure 2 inadvertently contained underlines and annotations due to a formatting issue. We have now properly exported and replaced the figure with a clean, high-quality version in the revised manuscript, ensuring there are no markings or corrections visible.

Comments 3: Lines 323-329: Although NRF2/KEAP1 signaling plays a key role is the studies discussed in this review, this signaling is poorly introduced. In fact, this pathway plays a key role in the progression and onset of several diseases (see PMID: 37296999, PMID: 39034715 ).

Response 3: Thank you for your insightful comment regarding the introduction of the NRF2/KEAP1 signaling pathway. In response, we have expanded the discussion in the revised manuscript to provide a more comprehensive overview of the pathway's role in oxidative stress regulation and its broader implications for various diseases, as noted in the suggested references (PMID: 37296999, PMID: 39034715). Specifically, we have clarified the mechanism by which NRF2 dissociates from KEAP1 during oxidative stress, its translocation to the nucleus, and the activation of antioxidant response elements (AREs), which in turn reduce reactive oxygen species (ROS) and mitigate cellular damage. This expanded section now underscores the significance of NRF2/KEAP1 signaling in ischemic stroke and highlights glycyrrhizic acid’s (GA) modulation of this pathway, including its impact on mitochondrial health, mitophagy, and neuroprotection. (See Lines 379-412)

Comments 4: A table summarizing the main results of the studies selected in this review should be reported

Response 4: Thank you for your valuable suggestion to include a table summarizing the main results of the studies in our review. In response to your feedback, we have added a comprehensive table in the revised manuscript that concisely outlines the key findings of each study discussed. We believe this addition enhances the clarity and accessibility of the review, as it allows readers to more easily compare and understand the therapeutic effects of glycyrrhizic acid across various studies. (See Table 1)

Comments 5: An accurate revision of typing errors is recommended

Response 5: Thank you for your attention to detail and for recommending a revision of typing errors. We have carefully proofread the entire manuscript and corrected any typographical errors we identified to enhance readability and precision. We appreciate your feedback, which has helped us improve the quality of our manuscript.

Round 2

Reviewer 4 Report

Comments and Suggestions for Authors

the manuscript has been significantly improved and can be accepted in the present form